# Electrochemical Sensor Based on Glassy Carbon Electrode Modified with Carbon Nanohorns (SWCNH) for Determination of Cr(VI) via Adsorptive Cathodic Stripping Voltammetry (AdCSV) in Tap Water

**DOI:** 10.3390/nano14171465

**Published:** 2024-09-09

**Authors:** Fabiana Liendo, Bryan Pichún, Amaya Paz de la Vega, Johisner Penagos, Núria Serrano, José Manuel Díaz-Cruz, Jaime Pizarro, Rodrigo Segura, María Jesús Aguirre

**Affiliations:** 1Departamento de Materiales, Facultad de Química y Biología, Universidad de Santiago de Chile, Santiago 9170022, Chile; 2Millennium Institute on Green Ammonia as Energy Vector—MIGA (ICN2021_023), Santiago 7820436, Chile; 3Department of Chemical Engineering and Analytical Chemistry, Universitat de Barcelona (UB), Martí i Franquès 1–11, 08028 Barcelona, Spain

**Keywords:** carbon nanohorns, modified electrode, chromium (VI), AdCSV

## Abstract

In this study, a new and simple glassy carbon electrode modified with carbon nanohorns (SWCNH/GCE) was used for the determination of Cr(VI) in aqueous matrices via adsorptive cathodic stripping voltammetry (AdCSV). The modified electrode was characterized via field emission scanning electron microscopy and cyclic voltammetry, which revealed a homogeneous distribution of spherical agglomerates of SWCNH on the electrode surface. The modification increased the electrochemically active area from 0.10 cm^2^ ± 0.01 (GCE) to 0.16 cm^2^ ± 0.01 (SWCNH/GCE). The optimized analytical conditions were as follows: a supporting electrolyte (0.15 mol L^−1^ HCl), an accumulation potential of 0.8 V versus Ag/AgCl, and an accumulation time of 240 s. Validation of the analytical methodology was performed, obtaining a linear range between 20 and 100 µg L^−1^, a limit of detection of 3.5 µg L^−1^, and a limit of quantification of 11.6 µg L^−1^ with good accuracy and precision. The method was applied to the analysis of spiked tap water samples, and the results were compared using a flame atomic absorption spectrophotometer (FAAS) with no significant statistical differences.

## 1. Introduction

During the last few decades, various anthropogenic and natural activities have caused an increase in the concentration of heavy metals in the environment. Some of them are toxic in all concentration ranges and others can accumulate and biomagnify along the food chain, which poses a serious problem for living organisms [1]. Chromium (Cr) is one of the most widely distributed heavy metals in the Earth’s crust [2], found mainly in trivalent (III) forms such as Cr(OH)_2_^+^ or Cr(OH)_3_ complexes and hexavalent (VI) forms such as oxo-compounds HCrO_4_^−^, CrO_4_^2−^, and Cr_2_O_7_^2−^ depending on the pH of the medium [3,4]. Cr(III) at low concentrations is an essential trace element for the normal and healthy development of living organisms. On the contrary, Cr(VI) is highly toxic in biological systems due to its strongly oxidizing nature. It can also enter cells through non-selective membrane channels, reducing to Cr(III) and generating reactive oxygen species (ROS), which cause DNA damage [3,5]. This leads to prolonged exposure to Cr(VI) with toxic, mutagenic, and carcinogenic effects on humans [3,6,7], this being classified as one of the most dangerous heavy metals [8]. Industrialization (textile, metallurgical, chemical, etc.) has led to an increase in exposure to Cr(VI) due to water and air pollution [8], causing it to enter human bodies through the ingestion of contaminated food and tap water, inhalation, or dermal contact [3]. For this reason, various entities such as the World Health Organization (WHO), Environmental Protection Agency (EPA), and Food and Drug Administration (FDA) restrict the concentration of total chromium in tap water, bottled water, and groundwater to 0.05, 0.10, and 50 mg L^−1^, respectively [3,8], and in the case of food to 1000 µg kg^−1^ [9].

It is evident that due to the toxic effects of Cr(VI), it is very important to verify compliance with regulations. Therefore, it is essential to have techniques that allow for its rapid quantification at the trace and ultra-trace levels. Among the main techniques used for the determination of total Cr are flame atomic absorption spectroscopy (FAAS), graphite furnace atomic absorption spectroscopy (GF-AAS) [10,11], inductively coupled plasma optical emission (ICP-OES) or mass spectrometry detection (ICP-MS) [1,12,13] and ion chromatography (IC) [14]. Although these techniques are highly sensitive, reproducible, and capable of measuring low concentrations, they involve high instrumentation and analysis costs, require trained personnel, and require significant sample preparation, which complicates routine analysis [15]. Additionally, in these techniques, selective detection of Cr(VI) poses a significant challenge due to interferences from Cr(III) [16]. As an alternative, electroanalytical techniques such as adsorptive cathodic stripping voltammetry (AdCSV) [17] represent fast detection systems with different advantages: simplicity, sensitivity, selectivity, reproducibility, and cost-effectiveness for detecting Cr at the trace and ultra-trace levels. In order to improve the voltammetric response of this type of system, various modified electrodes have been developed and reported in the literature, including polymers [17,18], metals [7,19], and carbon nanomaterials [20,21,22], among others [23,24]. Regarding carbon nanomaterials, single-walled carbon nanohorns (SWCNHs) have multiple advantages for the development of electrodic surfaces in addition to their low cost and eco-friendly character. SWCNHs are graphene sheets with conical horn-shaped tips and one-dimensional (1-D) structures, with diameters ranging from 2.0 to 5.0 nm and lengths between 40.0 and 50.0 nm [25]. They can assemble through weak Van der Waals interactions, forming spherical structures. These materials exhibit high electrical conductivity, a large specific surface area, an internal nanoporous structure, and no metallic residues, making them highly attractive for use as electrode modifiers in the electrochemical determination of heavy metals [22,25].

Different electrochemical sensors have been employed for the detection of Cr(VI) using voltammetric techniques. For example, Kachoosangi et al. used a carbon paste electrode (CPE), which they mixed with polyethylene and on which they deposited a gold film, obtaining a linear range between 20 and 2000 μg L^−1^ with a detection limit (LOD) of 4.4 μg L^−1^ [26]. Xu et al. employed a more complex sensor based on gold nanoparticles (AuNPs) functionalized with pyridine and reduced graphene oxide (RGO) (AuNPs/3D RGO/GCE), reporting a linear range of 25 to 300 µg L^−1^ and an LOD of 1.2 µg L^−1^ [27]. Filik et al. modified a screen-printed electrode (SPCE) with multi-walled carbon nanotubes (MWNCTs), neutral red (NR), and Au nanoparticles (MWCNTs-NR-AuNPs/SPCE) prepared via complex synthesis, which gave them a linear range between 21 and 4160 µg L^−1^ with an LOD of 1.3 µg L^−1^ [28]. Finally, a sensor reported by Sadeghi et al. stands out, as they used two-step modification with carbon nanotubes and quercetin (QH2) (QH2/MWCNT-SPCE) to obtain a sensor with a linear range between 52 and 10,400 µg L^−1^ and an LOD of 15.6 µg L^−1^ [29]. The reported studies focus on the determination of Cr(VI) in samples of tap water, river water, wastewater, and mineral water. This literature review also revealed that to the best of our knowledge there are no studies using working electrodes modified with carbon nanohorns for the detection of Cr(VI). Furthermore, the reported studies suggest the use of a difficult fabrication process for sensors with multiple modification steps, or in some cases, complex synthesis of the modifying agents [17,27,28,29], or the use of sensors that are not environmentally friendly [24]. It is also been observed that some sensors based on carbon nanomaterials exhibit linear ranges that start above those reported in this work, or in some cases, show higher LODs [26,27,29], which could represent a serious disadvantage compared to those obtained in this work.

To the best of our knowledge, the use of carbon nanohorns for chromium determination has not been previously reported. In this work, a simple, low-cost, and fast analytical methodology was developed for the direct determination of Cr(VI) in aqueous matrices using AdCSV with SWCNH/GCE. The sensor was characterized via cyclic voltammetry (CV) and field emission scanning electron microscopy (FE-SEM). Chemical and electrochemical parameters (HCl concentration, volume of the SWCNH suspension, E_acc_, and t_acc_) were optimized to achieve the best reduction in the voltammetric signal for Cr(VI). With the optimized parameters, the methodology was validated, demonstrating an excellent linear range and a low LOD, in contrast to those of other sensors fabricated from carbon nanomaterials. Finally, the methodology was successfully applied for the determination of Cr(VI) in tap water samples, yielding statistically comparable results to those obtained using atomic absorption spectroscopy.

## 2. Materials and Methods

### 2.1. Materials and Solutions

All reagents used were of analytical grade, including single-walled carbon nanohorns (SWCNH), N,N-Dimethylformamide (DMF), hydrochloric acid (HCl), potassium chloride (KCl), potassium ferricyanide trihydrate (K_3_[Fe(CN)_6_]·3H_2_O), potassium ferrocyanide trihydrate (K_4_[Fe(CN)_6_]·3H_2_O), potassium chloride (KCl) from Merck (Darmstadt, Germany), and potassium chromate (K_2_CrO_4_) from Fluka Analytical (Neu-Ulm, Germany). Solutions were prepared using ultrapure water obtained from a Milli-Q system (Burlington, MA, USA).

### 2.2. Apparatus

AdCSV and CV measurements were performed on a CH instruments 619E potentiostat (Austin, TX, USA). The measurements utilized a three-electrode system, with a modified glassy carbon electrode (SWCNH/GCE) as the working electrode, Ag/AgCl (3.0 mol L^−1^ KCl) as the reference electrode, and a Pt wire as the auxiliary electrode. Scanning electron microscopy (SEM) images were obtained using FE-SEM (ThermoFisher, Waltham, MA, USA, Quanta FEG 250). FAAS studies were performed on Analytik Jena (novAA 350, Jena, Germany).

### 2.3. Methods

#### 2.3.1. Preparation of SWCNH/GCE

A GCE was polished on a porous surface with 0.05 µm alumina and ultra-pure water. From a suspension of SWCNH (1.0 mg L^−1^) prepared in DMF and sonicated for 45 min, a 6.0 µL aliquot was taken and drop-casted on the electrode surface. The solvent was evaporated using infrared light for 60 min, and then the electrode was used to perform the corresponding measurements. The SWCNH/GCE surface was characterized via FE-SEM and CV using a redox probe (1.0 mmol L^−1^ [Fe(CN)_6_]^3−/4−^).

#### 2.3.2. Analytical Procedure for the Determination of Cr(VI)

The square-wave AdCSV technique was used for the determination of Cr(VI). Solutions of Cr(VI) were prepared in 0.15 mol L^−1^ of HCl as the supporting electrolyte. An accumulation potential (E_acc_) of 0.8 V was applied with an accumulation time (t_acc_) of 240 s. Cathodic scanning was performed in a potential range from 0.8 V to 0 V using the following parameters: potential increments of 4 mV, an amplitude of 25 mV, and a frequency of 15 Hz. All measurements were conducted at room temperature using SWCNH/GCE.

### 2.4. Sample Treatment

The tap water sample was collected in Santiago, Chile. It was filtered and spiked with 0.30 mg L^−1^ of Cr(VI) in 0.15 mol L^−1^ of HCl and used to prepare solutions that were measured in triplicate via the AdCSV technique using SWCNH/GCE. In addition, the spiked tap water samples were analyzed via FAAS.

## 3. Results and Discussion

### 3.1. Characterization of SWCNH and GC/SWCNH by SEM

Figure 1A shows the characteristic spherical assembly of SWCNH with an average diameter of 87 nm (Appendix A). The elemental analysis (from EDX) (Figure 1B) shows the predominant presence of C, without metallic impurities, since its synthesis was performed in the absence of metallic catalysts. SWCNH/GCE is presented in Figure 1C, showing the incorporation of SWCNH on the GC electrode surface with a relatively uniform distribution. As in the previous case, the EDX study showed that the incorporation of the nanomaterial is carried out without contaminating the electrode surface.

### 3.2. Electrochemical Characterization of the GCE and SWCNH/GCE

Figure 2A shows the CVs obtained by using 1.0 mmol L^−1^ of an [Fe(CN)_6_]^3−/4−^ redox probe at a scan rate of 0.1 V s^−1^ in a potential range from 0.5 to −0.1 V for GCE and SWCNH/GCE. A 1.5-fold increase in the cathodic and anodic peak currents was observed when using SWCNH/GCE as compared to those of GCE, which can be attributed to the increase in the number of active sites. Additionally, Figure 2 shows the CVs for GCE (Figure 2B) and SWCNH/GCE (Figure 2C) in 1.0 mmol L^−1^ of [Fe(CN)_6_]^3−/4−^ at different scan rates (from 0.02 to 0.12 V s^−1^). In the inset, the curves of current versus the square root of the scan rate (ν^1/2^) are displayed, which present linear behavior (R^2^ > 0.999), indicating that the process is controlled by diffusion. In this way, from the slope of these curves, it is possible to determine the electroactive area (A) using the Randles–Sevcik equation, Equation (1) [30], where D is the diffusion coefficient (7.6 × 10^−6^ cm^2^ s^−1^), C_o_ is the concentration of the redox probe (1.0 μmol mL^−1^), n is the number of electrons transferred (*n* = 1), and ν is the scan rate:I_p_ = 2.69 × 10^5^ nAC_0_ D^1/2^ ν^1/2^(1)

Using (1), the electroactive area for both electrodes was calculated using three replicates. Values of 0.10 ± 0.01 cm^2^ for GCE and 0.16 ± 0.01 cm^2^ for SWCNH/GCE were obtained. This increase in the electroactive area is attributed to the presence of SWCNHs, which possess a large surface area and, therefore, improve the electroactivity of the modified electrode.

### 3.3. Electrochemical Determination of Cr(VI)

#### 3.3.1. Supporting Electrolyte Study

The chemical equilibrium of Cr(VI) is dependent upon the pH of the medium. At low Cr concentrations and at an acidic pH, the dominant species are HCrO_4_^−^ and Cr_2_O_7_^2−^, of which the electrochemically active species is the HCrO_4_^−^ anion [31]. Sasithorn [17], Chavez-Lara [32], and Sadeghi [29], among others, have reported the use of HCl as a supporting electrolyte for the determination of Cr(VI). The use of HCl results in the formation of both HCrO_4_^−^ and CrO_3_Cl^−^ complexes. The latter has no electrochemical activity, and thus it is crucial to assess the impact of varying HCl concentrations on the formation of HCrO_4_^−^. The effect of HCl concentration on the current intensity obtained for a reduction of 100 µg L^−1^ Cr(VI) was evaluated in the range of 0.05 to 0.25 mol L^−1^ with E_acc_ 0.8 V and t_acc_ 240 s (Figure 3A). An increase in current was observed as the HCl concentration increased up to 0.15 mol L^−1^, gradually decreasing at higher concentrations. When the HCl concentration was 0.15 mol L^−1^ the dominant species in the solution was HCrO_4_^−^, forming less than 10% of the chromium chloride complexes [31]. Accordingly, 0.15 mol L^−1^ of HCl was selected as the optimal concentration for subsequent investigation.

#### 3.3.2. Effect of the Volume of SWCNH Suspension

The effect of the aliquot volume of the SWCNH suspension (1.0 mg mL^−1^) for the modification of the GCE on the cathodic peak current was evaluated using 100 µg L^−1^ of Cr (VI) and aliquot volumes between 0 and 10.0 µL of the SWCNH suspension (E_acc_ 0.8 V, t_acc_ 120 s) (Figure 3B). An increase in cathodic current was observed with the increasing SWCNH aliquot volume, reaching a maximum at 6.0 µL and decreasing at higher volumes. When using an aliquot of 6.0 µL of SWCNH, better coating of the GCE surface was achieved, causing an increase in the active sites, whereas at higher volumes, multilayers were generated on the electrode surface, hindering electronic transfer and decreasing the current response. For this reason, an aliquot of 6.0 µL of the SWCNH suspension was used to fabricate the SWCNH/GCE.

#### 3.3.3. Effect of Potential and Time of Accumulation

The effect of t_acc_ on the cathodic peak current using 100 µg L^−1^ of Cr (VI) is shown in Figure 3C, with t_acc_ ranging from 0 to 420 s and an E_acc_ of 0.8 V, as previously reported [17,32]. An increase in the reduction current was observed as the t_acc_ increases, reaching a maximum at 240 s. After this time, no further increase was observed, which is attributed to the saturation of the active sites. The optimal t_acc_ was selected to be 240 s to measure concentrations lower than 100 µg L^−1^ of Cr.

The E_acc_ was evaluated between 0.6 and 1.0 V using 100 µg L^−1^ of Cr(VI) and a t_acc_ of 240 s (Figure 3D). An increase in the reduction current was observed as the E_acc_ increased from 0.6 V to 0.8 V and decreased thereafter. Different authors such as Chavez-Lara et al. [32] and Phetlada Sanchayanukun et al. [17] explain that at 0.8 V (vs. Ag/AgCl), Cr(VI) adsorption occurs without electrochemical reduction, and upon the application of cathodic scanning, Cr(VI) to Cr(III) reduction occurs, as shown in (2) [32]. Therefore, 0.8 V was chosen as the optimum E_acc_.
(2)$$HCrO_{4}^{-}+3e^{-}\to Cr^{3+}$$

#### 3.3.4. Voltammetric Response of Cr(VI)

Using the optimal parameters, the voltammetric response of 100 µg L^−1^ of Cr(VI) solution obtained using SWCNH/GCE was compared with that of the GCE using AdCSV (Figure 4). The results indicate that the SWCNH/GCE provides a current signal of −4.40 µA in response to the presence of Cr(VI) at 0.590 V, while no signal is observed with the bare GCE.

### 3.4. Analytical Validation and Applications

#### 3.4.1. Calibration Curve, Limit of Detection, and Quantification

The analytical methodology was validated through a study of the linear range, the limit of detection and quantification (LOD and LOQ), reproducibility, repeatability, selectivity, and accuracy, using the SWCNH/GCE at the optimal parameters: HCl 0.15 mol L^−1^, E_acc_ 0.8 V, and t_acc_ 240 s. Once validated, the method was applied for the determination of Cr(VI) in tap water samples.

Figure 5A presents voltammograms and a calibration curve, which shows a linear increase in the cathodic peak current with increasing Cr(VI) concentration. A linear range between 20 and 100 µg L^−1^ Cr(VI) was obtained, with a straight-line equation of y = 0.0649x − 0.461 and a correlation coefficient of 0.999. The LOD and LOQ were calculated to be 3 and 10 times the standard deviation of the calibration curve from the slope [33,34], giving values of 3.5 µg L^−1^ and 11.6 µg L^−1^, respectively.

Table 1 presents the linear ranges and LODs from some of the reported works using carbon-nanomaterial-modified working electrodes for the determination of Cr(VI) using different voltammetric techniques. In most of the cases, complex electrode fabrication with several modification steps or the use of complex syntheses to obtain the modifier [27,28,29,32,35] is reported, where these proposed modifications do not provide linear ranges and/or LODs [26,27,28,29,36] better than those obtained in this work. In this context, it is possible to indicate that the SWCNH/GCE is highly competitive due to the simplicity of its fabrication, with analysis and LOD ranges similar to those obtained with electrodes based on carbon materials of more complex fabrication.

#### 3.4.2. Repeatability, Reproducibility, and Effect of Interferents

Repeatability was evaluated by studying the current intensity obtained with the same electrode when analyzing 10 times the current of the reduction peak for 100 µg L^−1^ of Cr(VI), obtaining a relative standard deviation (RSD) of 6.5%, which indicates that the electrode is stable in the range of the measurements performed (Figure 5B). The reproducibility was studied by means of the evaluation of the current intensity obtained for 10 different electrodes. These measurements present an RSD of 9.0%, which indicates that the developed sensor presents good reproducibility (Figure 5C). The findings suggest that the SWCNH/GCE can be utilized for a minimum of 10 consecutive times without compromising the intensity of the current obtained, and it is reproducible for the determination of Cr(VI).

On the other hand, in order to investigate whether other species present in a solution affect the intensity of the cathodic peak obtained for the determination of Cr(VI), selectivity was evaluated. The possible interference of some cations was studied individually via the addition of 10 and 100 µg L^−1^ of Fe(III), Cr(III), Mg(II), Zn(II), Ca(II), Cu(II), Co(II), Pb(II), Ni(II), Na(I), and K(I) to a solution of 50 µg L^−1^ of Cr(VI). The effect of the interferents was evaluated by calculating the variation in the current intensity, defining an error less than or equal to 10% as the tolerance limit [17]. Figure 5D shows that, for most of the cases, at low concentrations of the interferents (10 µg L^−1^), the signal obtained for Cr(VI) was not significantly modified, with a variation in the current greater than 10% only when Ca, Pb, and Na were used. However, when a concentration of 100 µg L^−1^ was used, a variation in the current greater than 10% was observed in all cases. The effect of the interferents on the Cr(VI) signal could be attributed to the competition for active sites, as a decrease in current intensity was observed. Cr(VI) quantification can be performed using the standard addition method, so the effect on the Cr(VI) signal caused by the presence of the different cations studied will be compensated for by successive additions of Cr(VI) standards, allowing its quantification even in the presence of interfering species [17,20].

#### 3.4.3. Accuracy

The accuracy of the method was evaluated by determining the amount of Cr(VI) in the spiked tap water samples using the standard addition method. The tap water sample was spiked with a known concentration of the Cr(VI) standard, from which 10 mL of a solution of 20.0 μg L^−1^ and 30.0 μg L^−1^ of Cr(VI) in 0.15 mol L^−1^ of HCl was prepared. Three independent analyses (*n* = 3) of both spiked tap water samples were performed using three SWCNH/GCE, obtaining recovery percentages of 105% and 110%, respectively, as shown in Table 2. According to Association of Official Analytical Chemists International (AOAC International) criteria, the proposed methodology is considered accurate for the determination of Cr(VI) in tap water samples, and it is possible to indicate that the signal obtained for Cr(VI) does not experience significant interferences due to the matrix effect.

Therefore, the validation of the proposed analytical methodology indicates that the SWCNH/GCE can be used for the determination of Cr(VI) in real water samples.

#### 3.4.4. Real Sample Analysis

Tap water samples were analyzed via AdCSV using the SWCNH/GCE and FAAS. In both techniques, the presence of Cr(VI) was not detected, probably because its concentration was below the LOD; therefore, the spiked tap water samples were analyzed. AdCSV analysis was performed in triplicate, using a different SWCNH/GCE each time and employing the standard addition method with optimized parameters (HCl 0.15 mol L^−1^, E_acc_ 0.8 V, and t_acc_ 240 s). The results obtained were compared with those obtained using the FAAS technique.

In the analysis of the spiked tap water sample via AdCSV using the SWCNH/GCE and FAAS, concentrations of 33.0 ± 2.4 μg L^−1^ and 31.4 ± 0.7 μg L^−1^ of Cr(VI) were detected, respectively (Table 3). The analysis of the results using Student’s *t*-test [17,37] (for the comparison of replicate measurements) shows that there is no significant statistical difference between the measurements obtained via both methods since the calculated t-value was less than the critical t-value (4.3) for two degrees of freedom (*p* = 0.05). This allows us to conclude that it is possible to use the SWCNH/GCE in AdCSV technique for the accurate determination of the amount of Cr(VI) in water samples and that there is no significant interference of the matrix effect when analyzing the samples. The results obtained indicate that the developed method is accurate and reliable for the determination of Cr(VI) in real water samples.

The developed method proposes the use of a sensitive electrode of simple manufacture, without the need to use complexing substances for the detection of Cr(VI), obtaining competitive detection limits comparable to those reported in other works (Table 1).

## 4. Conclusions

In this work, it was possible to successfully implement a validated methodology for the determination of Cr(VI) via AdCSV in tap water samples, using a working electrode fabricated in a simple way using the drop coating technique. The SWCNH/GCE surface was characterized via SEM, showing the uniform incorporation of the nanomaterial on the electrode surface and the characteristic spherical assembly of SWCNH. In addition, the calculation of the electroactive area via CV showed an increase of 60% as compared to that of the GCE. The electrochemical parameters were optimized, obtaining a better current response when using 0.15 mol L^−1^ of HCl as the supporting electrolyte and E_acc_ and t_acc_ values of 0.8 V and 240 s, respectively. Finally, validation of the analytical methodology was performed, obtaining a linear range between 20 and 100 µg L^−1^, an LOD of 3.5 µg L^−1^, and an LOQ of 11.6 µg L^−1^ with good accuracy (105 and 110% recovery). Real samples were analyzed using AdCSV with the optimal parameters via standard addition method, and the results were compared with those of FASS, obtaining statistically comparable results. Thus, it was demonstrated that the SWCNH/GCE could be applied in the determination of the amount of Cr(VI) via AdCSV in real samples in a precise and accurate way.

## Figures and Tables

**Figure 1 nanomaterials-14-01465-f001:**
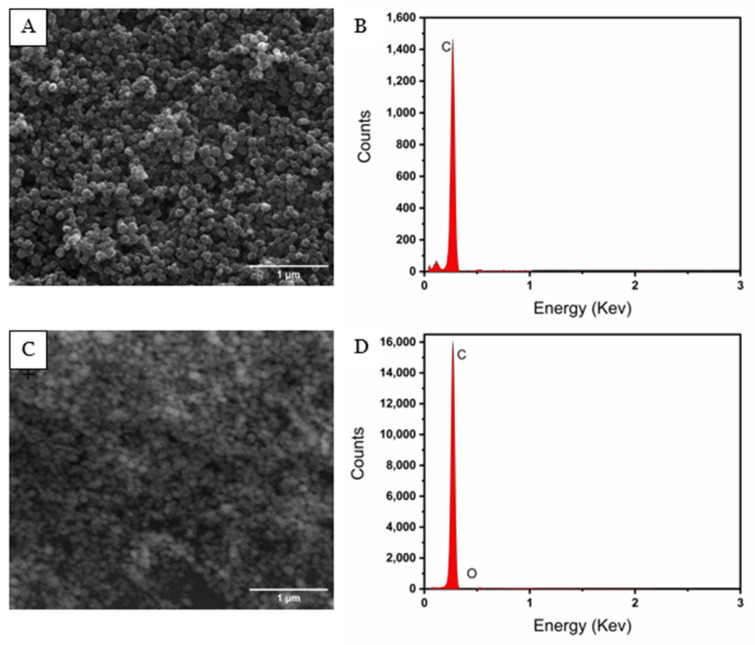
FE-SEM images of (**A**) SWCNH and (**C**) SWCNH/GCE. EDX images of (**B**) SWCNH and (**D**) SWCNH/GCE.

**Figure 2 nanomaterials-14-01465-f002:**
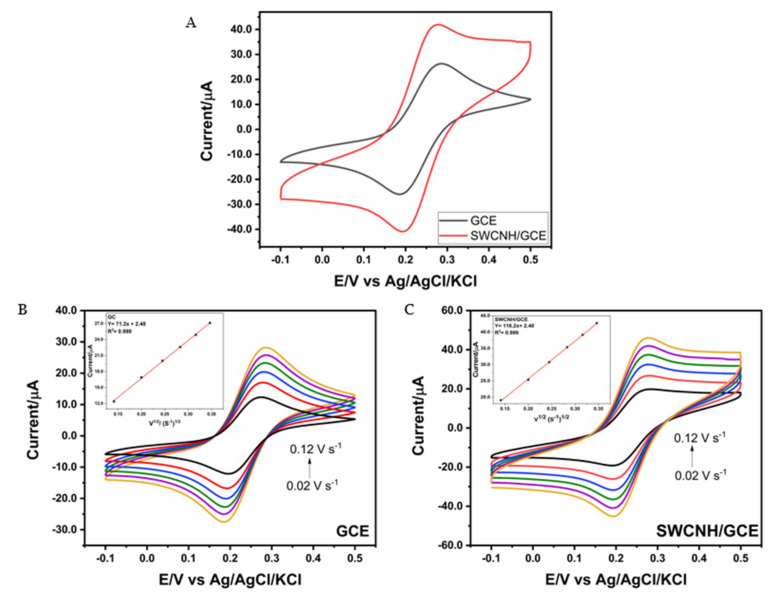
(**A**) Cyclic voltammograms of GCE and SWCNH/GCE in presence of 1.0 mmol L^−1^ [Fe(CN)_6_]^3−/4−^ at 0.1 V s^−1^. (**B,C**) Cyclic voltammograms of GCE and SWCNH/GCE at different scan rates (0.02–0.12 V s^−1^) in [Fe(CN)_6_]^3−/4−^ (1.0 mmol L^−1^) (inset: current vs. square root of scan rate).

**Figure 3 nanomaterials-14-01465-f003:**
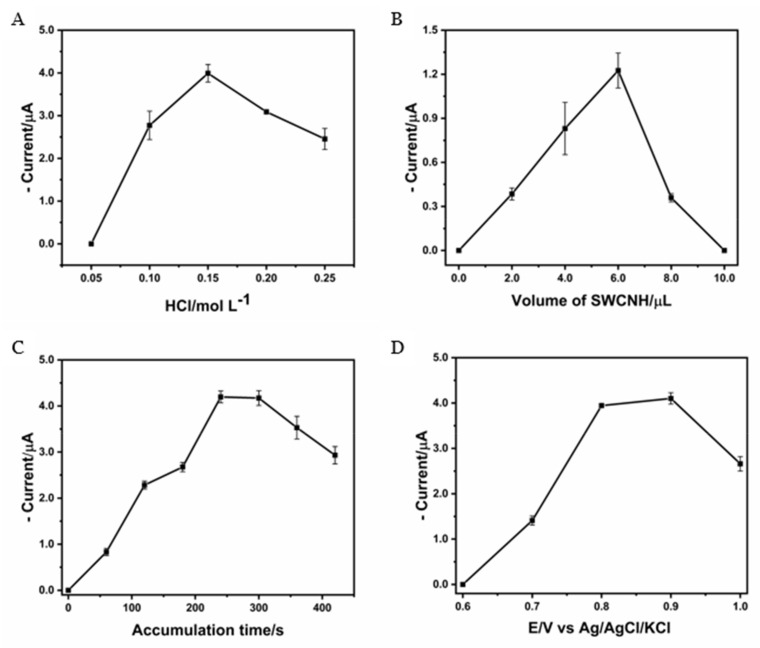
Current plots as a function of (**A**) HCl concentration, (**B**) SWCNH suspension volume, (**C**) accumulation time, and (**D**) accumulation potential, using SWCNH/GCE.

**Figure 4 nanomaterials-14-01465-f004:**
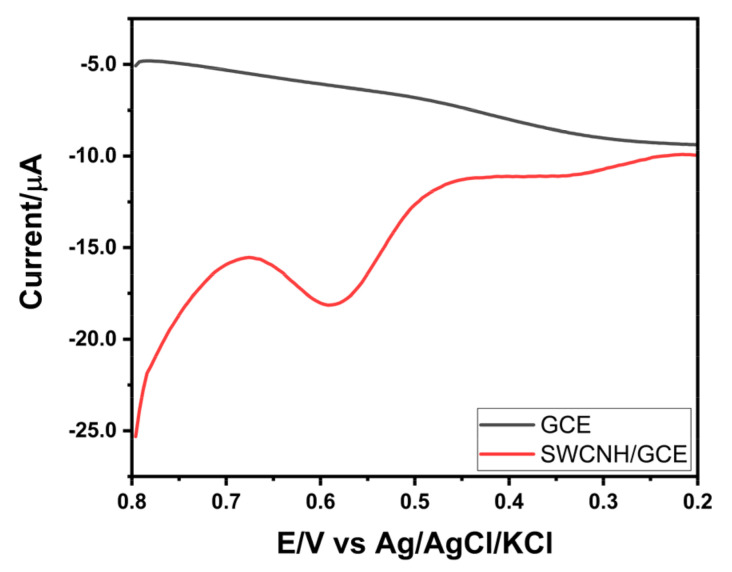
Electrochemical behavior of 100 µg L^−1^ of Cr(VI) in GCE and SWCNH/GCE, using the optimized parameters (HCl 0.15 mol L^−1^, E_acc_ 0.8 V, and t_acc_ 240 s).

**Figure 5 nanomaterials-14-01465-f005:**
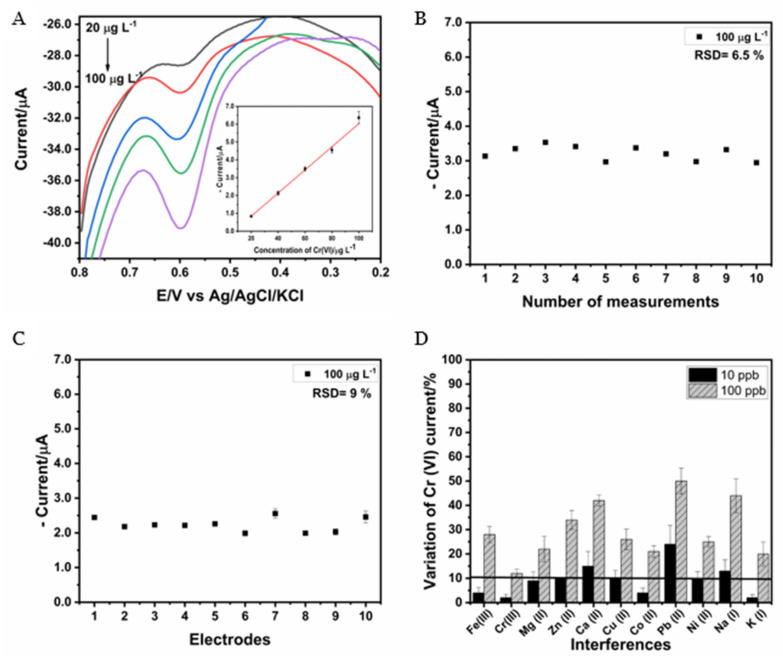
(**A**) Voltammograms and calibration curve for Cr(VI), (**B**) repeatability study, (**C**) reproducibility analysis, and (**D**) effect of interferents using SWCNH/GCE under optimized conditions. HCl 0.15 mol L^−1^, E_acc_ 0.8 V, and t_acc_ 240 s.

**Table 1 nanomaterials-14-01465-t001:** Comparison of the analytical performance of the SWCNH/GCE with some of the electrodes reported in the literature modified with carbon nanocomposites for the detection of Cr(VI) using electrochemical techniques.

Electrode	Technique	μg L^−1^		Sample	Ref.
		Lineal Range	LOD		
Film of Au/polyethylene-technical carbon	LSV	20 to 2000	4.4	River water	[26]
Pyridine functionalized AuNPs/3D RGO/GCE	AdSV	25 to 300	1.2	Waste water	[27]
MWCNTs-NR-AuNPs/SPCE	LSV	21 to 4160	1.3	Tea, milk and mineral water	[28]
QH2/MWCNT/SPCE	DPV	52 to 10,400	15.6	Tap, Mineral and River water	[29]
AuNP/MWCNT-Chit/GCE	DPV	0.003 to 0.1	0.007	Tap water	[32]
rGO/NiS/AuNCs/GCE	SWASV	2 to 14	0.09	Ground water	[35]
T-GO-C/GCE	SWV	5 to 600	20	Tap and tannery water	[36]
SWCNH/GCE	AdCSV	20 to 100	3.5	Tap water	This work

LSV: linear sweep voltammetry. AdSV: adsorptive stripping voltammetry. DPV: differential pulse voltammetry. SWASV: square-wave anodic stripping voltammetry. SWV: square-wave voltammetry. Chit: chitosan. NiS: nickel sulfide. AuNCs: gold nanocube. T-GO-C: Thymine-GO-Carbohydrazide.

**Table 2 nanomaterials-14-01465-t002:** Determination of Cr(VI) in tap water samples (*n* = 3) via AdCSV technique according to optimized methodology.

Sample	Cr(VI) Concentration (µg L^−1^)		Recovery (%)
	Added	Measurement	
Spiked sample	20.0	21.0 ± 1.8	105
	30.0	33.0 ± 2.4	110

**Table 3 nanomaterials-14-01465-t003:** Determination of Cr(VI) in spiked tap water (*n* = 3) via FAAS and AdCSV.

Samples	FAAS	AdCSV	*t* de Student
	(μg L^−1^)		
Tap water	ND	ND	-
Spiked tap water	31.4 ± 0.7	33.0 ± 2.4	1.09

ND = not detected.

## Data Availability

The data are presented within the article.

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
