# Peer review of "Electrochemical Sensor Based on Glassy Carbon Electrode Modified with Carbon Nanohorns (SWCNH) for Determination of Cr(VI) via Adsorptive Cathodic Stripping Voltammetry (AdCSV) in Tap Water"

_nanomaterials, 2024, doi:10.3390/nano14171465_

Round 1

Reviewer 1 Report

Comments and Suggestions for Authors

The authors presented a simple and fast method for the detection of Cr(VI).

The manuscript presents good English level, with some small mistakes and some unclear phrases identified in the text (see attached document). It should be corrected.

The study was well organized and described, with some possible improvements. The manuscript should better underline the real samples analysis and selectivity.

In the attached document you have my remarks and suggestions for improving the manuscript. For these reasons, I do recommend the publication of this manuscript after minor revision.

Reviewer 2 Report

Comments and Suggestions for Authors

In this study, the author proposes a novel analytical method for Cr(VI) in aqueous matrices using AdCSV with SWCNH/GCE. Finally, the electrochemical sensors are applied to tap water samples containing Cr(VI). However, some questions need to be corrected. The questions are as follows:

1. In the study of repeatability, reproducibility analysis, and the effect of interferents, the fig5 is short of error bars.

2. About real sample analysis, the author should compare with other electrochemical detectors to prove the advantage of SWCNH/GCE.

3. Compared with other electrochemical detectors modified with SWCNH, what is the reason for the advantages of this paper in detection?

4. The particle size characterization of SWCNH should be supplemented.

Comments on the Quality of English Language

The English expression is relatively fluent and accurate.

Reviewer 3 Report

Comments and Suggestions for Authors

In this manuscript, the authors developed a single-walled carbon nanohorns (SWCNHs) modified GCE electrode for electrochemical sensing of Cr(VI). This work is generally systematic, and the authors present a comprehensive optimization of the analytical method. However, my main concern about this manuscript is that it lacks the novelty required by Nanomaterials. Various carbon materials like MWCNTs and GOs have already been utilized for the electrochemical analysis of Cr(VI) in existing literature. The purported advantages of SWCNHs, such as a large surface area, internal nanoporous structure, and absence of metallic residue, are also shared by other carbon nanomaterials. As shown in Table 1, the sensing performance of the electrode in this work does not stand out compared to previously reported results. Therefore, I suggest that a major revision is necessary before considering this manuscript for publication.

Specific comments:

1. In addition to SEM, more characterization data should be provided to support the successful synthesis of SWCNHs.

2. The authors utilized the technique of adsorptive cathodic stripping voltammetry for the analysis of Cr(VI). Given that this technique is not widely employed in the literature and necessitates a longer analytical duration, it is essential to emphasize its advantages over LSV and DPV.

Round 2

Reviewer 3 Report

Comments and Suggestions for Authors

Although the manuscript has been revised according to some of the suggestions of the reviewers, my concern about the novelty and value of this manuscript, which is given in the first paragraph of my previous report, is not addressed. Therefore, further revision is needed before the publication of this manuscript. 

Round 3

Reviewer 3 Report

Comments and Suggestions for Authors

All the concerns have been properly addressed, and this manuscript could be published now.